# Experimental Investigation on the Mechanical Characteristics of Cement-Based Mortar Containing Nano-Silica, Micro-Silica, and PVA Fiber

Hossein Nematian Jelodar [1], Ata Hojatkashani [1,2,*], Rahmat Madandoust [3], Abbas Akbarpour [1] and Seyed Azim Hosseini [1]

[1] Department of Civil Engineering, Islamic Azad University, South Tehran Branch, Tehran 1584743311, Iran
[2] Nanotechnology Research Center, Islamic Azad University, South Tehran Branch, Tehran 1584743311, Iran
[3] Department of Civil Engineering, University of Guilan, Rasht 1376941996, Iran
* Correspondence: ata_hojat@aut.ac.ir

**Abstract:** This paper investigates bending and compressive strengths as mechanical characteristics of cement-based repair mortar containing nano-silica (NS) and micro-silica (SF) as cement replacements particles and polyvinyl alcohol (PVA) fibers. The mentioned materials were added to the mortar in three different conditions, including single (just one material), binary (mixture of two admixtures), and ternary (mixture of all three admixtures) modes. The use of PVA fibers, nano-silica and micro-silica in the triple combination of a cement-based mortar is the primary objective of the current research. In total, 28 mix designs with various percentages of particles and fiber were employed in the current study, and 112 different specimens were prepared to conduct the experimental research. The compressive and flexural strength results have been selected as the criteria for obtaining the optimum mix design for each condition. In order to specify the mechanical characteristics of specimens, a compressive test was carried out according to ACI 318, and the three-point bending test was utilized according to BS EN 1015-11. The results obtained from this study show that the mixture containing 10% silica fume (SF10) can be considered the optimum mix design for the single-mode condition. For such a mix design, a flexural strength increase of 27% and a compressive strength improvement of 48% were determined in comparison to the reference mixture design. The mixture containing nano-silica at 2% and silica fume at 8% (NS2SF8) was the optimum mix design in the binary mode condition. With the current mix design, a flexural strength improvement of 24% and a compressive strength increase of 49% in a 28-day specimen were recorded. Finally, under the ternary mode condition, a flexural strength enhancement of 3.5% and a compressive strength improvement of 4.6% were obtained. Additionally, the mixture design containing a PVA content of 0.75% and an SF content of 10% (PVA0.75SF10) was considered optimum.

**Keywords:** cement-based mortar; nano-silica; micro-silica; PVA fiber; flexural strength; compressive strength; modulus of rupture

## 1. Introduction

The use of cement-based mortars is prevalent in repairing and reconstructing deteriorated concrete and masonry structures that have shown signs of aging and considerable damage due to severe climatic conditions and mechanical factors, such as external loads [1]. There is a pressing need to utilize repair mortars for two main reasons, including the deterioration and reduction in the intended service life of structures and to secure safety and serviceability. An acceptable repairing material enhances structural function and performance and recovers and increases strength and durability [2]. The principal constituents of repair mortars are sand, binder, and water, the specifications of which could be modified by incorporating additional materials such as PVA and nano-silica [3]. Cement-based mortars are the most widely used elements among repair materials consisting of conventional

cement mortar, and they often incorporate one or more additives, such as nano-silica, silica fume, or other industrial byproducts [2].

Nanoparticles have attracted the attention of the construction industry and academic sectors due to their distinctive behaviors. The development of mortars with a high compressive strength requires higher cement content, which plays a crucial role in the mechanical characteristics and can lead to high concrete production and carbon emissions with destructive environmental effects. Minerals such as nano-silica and silica fume have been added to concrete to eliminate such disadvantages. It has been shown that the partial usage of mineral admixtures instead of cement in concrete can improve its properties, such as mechanical characteristics and thermal stability [4]. Several papers have been published in recent years regarding the utilization of nanoparticles in cement-based mortars [5]. According to previous investigations, the compressive strength can be increased as the nano-silica weight percentage increases [6–9]. Jo et al. concluded that the compressive strength of mortars containing nano-silica is higher than that of mortars containing silica fume after 7 and 28 days [10]. Sanchez and Sobolev conducted comprehensive research utilizing nanoparticles and their combinations in the mix design of cement mortars. Some other studies have shown that nanoparticles have a great impact on the strength and durability of mortars [11]. Sumesh et al. investigated the effect of using nano-structure materials, such as nano-silica, nano-copper and nano-ferrous, on the engineering properties of cement mortar. The results show that the incorporation of nano-materials in both cement composites and geo-polymers had better performance effects compared to the samples without nano-materials, which is mainly attributed to the combined effects of the pozzolanic activity and filler effects of nano-materials forming denser microstructure [12]. Zapata et al. evaluated the effects of micro-silica and nano-silica at various dosages on the engineering properties of fresh and hardened mortars. Better fluidity and, hence, better mortar compactions were attained with the micro- and nano-$SiO_2$ additions [13]. Oltulu and Sahin assessed the effects of incorporating nano-Silica, nano-$Al_2O_3$, and nano-$Fe_2O_3$ on the engineering properties of cement mortars by comparing their compressive strengths and capillary water absorption capacities. For each mortar, increases in the compressive strength and decreases in the capillary absorption were determined relative to the control specimen [14]. Adhikary et al. (2021) investigated the effects of carbon nanotubes on the properties of lightweight aggregate concrete containing expanded glass and silica aerogel. The addition of carbon nanotubes has been shown to achieve good improvements in compressive strength. SEM images of lightweight concrete showed a homogeneous dispersal of carbon nanotubes within the concrete structure [15]. Enhanced durability properties have been attained by the pore filling and nucleation effect of CNTs, which, in turn, reduces the number of micro-pores and nano-pores [16].

Researchers have studied the effects of various types of fibers on the performance of cementitious materials. In an experimental study, the effects of different types of polymers on the mechanical properties of polymer-modified cement mortars (PCMs) were investigated [17]. In one of the latest works, the relationships between the mechanical properties and loading speed of polypropylene fiber (PPF)-incorporated cement mortar at different ages were studied [18].

Fiber-reinforced mortar (FRM) is a promising material from the point of view of its high energy absorption ability [19] and its ability to avoid multiple crack distribution [20]. Therefore, FRM presents a pseudo-ductile behavior, maintaining a considerable load-carrying capacity after matrix cracking with reduced crack widths. Thus, despite a slight degradation in the compressive strength, the flexural strength and fracture toughness can be significantly improved due to the presence of fibers [21–23]. Polyvinyl alcohol (PVA) is one of the best materials in this context and is becoming increasingly attractive as it has a higher shear and splitting strength, and a cheaper production cost, than other types of fibers [24]. PVA is a synthetic polymer with high tensile strength, elastic modulus, and durability; it is also environmentally friendly and can be totally decomposed in the environment in a short time without causing any pollution [25,26]. The PVA FRM was

found to have a higher tensile strength capacity [27] and higher workability with lower compressive strength [28–32] compared to the control mortar. In addition, the PVA fiber showed proper endurance in a cement environment with a hydrophilic surface, whereby strong chemical compounds can be made with a cement matrix [33]. Adhikary et al. (2019) showed that polyolefin had a significant influence on the concrete's strength and self-compaction properties. Higher doses of polyolefin provided better flexural strength. The post-cracking behavior was also improved by the addition of fibers. The compressive strength of concrete has also been enhanced by the addition of fibers [34]. Regarding the interesting characteristics of nano-materials, such as nano-alumina, nano-silica, nano-titania and MWCNTs, on the mechanical behavior of cement mortar, researchers investigated the effects of the combination of such materials on enhancing both the compressive and flexural strengths of the cement paste [35,36].

The combination of nanoparticles and fiber in mortars has become an interesting topic among scholars in recent years. In this approach, concrete can be made greener by using nano-materials instead of cement, which reduces its environmental impact via a reduction in $CO_2$ emissions in cement production [37]. On the other hand, a better performance and more ductile behavior can be achieved by adding fibers to the mortar to provide improved specifications, such as enhanced energy absorption capacity with a lower crack-propagation rate and a smaller crack width after matrix cracking. Mohseni et al. investigated the effects of incorporating rice husk ash (RHA) and nano-alumina (NA) as supplementary materials and polypropylene fiber as the reinforcement in cement mortar. Some engineering parameters, such as the compressive and flexural strengths, water absorption and drying shrinkage of hardened composites, have been evaluated, and the positive effect of adding these materials has been shown [37].

This study is an experimental program aimed at investigating the effects of the combination of nano-silica, micro-silica and PVA fibers. Such materials have been utilized to carry out experimental tests in various mix designs to achieve the optimum mixture (regarding the improvement of mechanical characteristics) under three different conditions, including single, binary and ternary modes, using the materials mentioned previously. The compressive and flexural strength parameters have been selected as the decisive parameters in specifying optimized mixtures in each mode, and some other valuable results have been obtained.

## 2. Experimental Procedure

### 2.1. Materials Physical Properties

Portland cement (Type 1) and micro-silica (silica fume) were used in this study. The 28-day minimum compressive strength of this type of cement is 42.5 MPa. Physical properties of such materials are presented in Table 1.

**Table 1.** Physical properties of cement and silica fume (SF).

| Physical Properties | Cement 425–1 | Micro-Silica |
|---|---|---|
| Specific gravity (kg/m$^3$) | 3090 | 2160 |
| Fineness (cm$^2$/kg) | 3,299,000 | 199,980,000 |

Naturally, a change in any of the variables, materials and conditions of composition and testing can affect the strength and mechanical properties of mortars. In this research, in order to properly investigate the effects of using additives, an attempt has been made to maintain the other specifications as constant as possible. For this reason, only type-1 cement was used for the compounds. The PH was measured as 8.5.

Natural silica sand with round corners without any organic impurities was the reference sand utilized in this investigation, for which the specified particle size distribution was in accordance with the BS EN 196-1 (2016) [38]. The nano-silica used in this research was

the colloid nano-silica with a +98% purity. Physical and chemical properties of nano-silica are reported in Table 2.

**Table 2.** Physical and chemical properties of consumed nano-silica.

| Dimensions (nm) | Specific Surface Area (m²/kg) | Solid Content (*w/w*) | Density (kg/m³) | Purity | Viscosity (Pa.m.s) | PH |
|---|---|---|---|---|---|---|
| 15 | 170,000 | 18% | 1120 | >98% | <10 | 8.5 |

A third-generation superplasticizer based on modified poly-carboxylate ether with the trade name of PS-10 was used due to the presence of pozzolans, particularly nano-silica, in the composition, with high fineness and a high specific surface area, in order to fix the mixture's consistency, while the water to cement ratio was kept constant.

Polyvinyl alcohol (PVA) fiber was utilized in the cement compositions due to its high formability and energy absorption capability [39]. The specifications of the materials, including fibers, superplasticizers, cement, etc., are based on the specifications used in the factory, and the water used was the drinking water from Tehran province. Additionally, the dimension of the sample used to determine the compressive strength of the cement mortars was 40 mm. The shape and specifications of PVA fiber are presented in Figure 1 and Table 3, respectively.

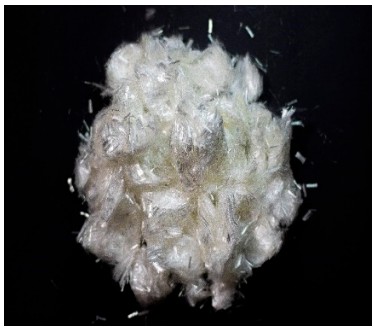

**Figure 1.** PVA fiber.

**Table 3.** Specifications of PVA fiber.

| Specific Weight (kg/m³) | Final Strain (%) | Elastic Modulus (GPa) | Tensile Strength (MPa) | Length (mm) | Diameter (μm) |
|---|---|---|---|---|---|
| 1300 | 7 | 40 | 1600 | 6 | 16 |

### 2.2. Control Specimen and Mixture Design

At first, the control mix design was determined based on the principal constituents, including sand, cement, and water, according to the weight ratios of the BS EN 196-1 standard [38]. The entire mix design used for all specimens has been categorized into seven different schemes. For Scheme A, in which nano-silica (NS) was included, the mixture with the highest bending and compressive strengths was the optimized mixture (Aopt). For scheme B, wherein the micro-silica (SF) was incorporated, the optimized mixture is referred to as Bopt. Scheme C contained nano-silica and micro-silica (NS + SF), and the optimized compound was called Copt. Scheme D includes the PVA filaments with various volume percentages (0.25 to 0.75), and the optimized mix design is named Dopt. For the E and F schemes, mixtures of PVA filaments with Aopt and Bopt were investigated, respectively, and the seventh scheme (G) contained a mixture of PVA with Copt.

In Schemes D, E, F, and G, three parameters were involved in selecting the optimized mixture, which are maximum bending strength, maximum compressive strength, and the

bridging phenomenon between cracks due to the existence of PVA fiber. Eventually, three mix designs (one from each mode) among the seven mentioned schemes were selected as the optimized mixtures with the highest strengths in relation to the bridging phenomenon between the cracks. All the mentioned mixtures are outlined in Table 4.

**Table 4.** Percentages of nano-silica, micro-silica, PVA, and superplasticizer (SP).

| Row | Scheme | Constituent | | | | % | | |
|-----|--------|-------------|------|------|------|------|------|------|
| 1 | A | NS | NS | 1 | 2 | 4 | 6 | 8 | 10 |
| | | | SP | 0.53 | 0.7 | 1.12 | 1.67 | 2.44 | 3.56 |
| 2 | B | SF | SF | 1 | 2 | 4 | 6 | 8 | 10 |
| | | | SP | 0.18 | 0.27 | 0.31 | 0.33 | 0.36 | 0.38 |
| 3 | C | NAS +SF | NS + SF | NS1SF9 | NS2SF8 | NS3SF7 | NS4SF6 | NS5SF5 | NS6SF4 |
| | | | SP | 0.66 | 0.71 | 0.91 | 1.35 | 1.89 | 2.47 |
| 4 | D | PVA | PVA | 0.25 | | 0.5 | | 0.75 | |
| | | | SP | 0.44 | | 1.78 | | 3.31 | |
| 5 | E | PVA + NS | PVA + NS | PVA0. 25Aopt | | PVA0. 5$A_{opt}$ | | PVA0.75$A_{opt}$ | |
| | | | SP | 1.26 | | 2.45 | | 3.89 | |
| 6 | F | PVA + SF | PVA + SF | PVA0. 25Bopt | | PVA0. 5$B_{opt}$ | | PVA0.75$B_{opt}$ | |
| | | | SP | 2.21 | | 3.33 | | 3.78 | |
| 7 | G | PVA + NS + SF | PVA + NSSF | PVA0. 25Copt | | PVA0. 5$C_{opt}$ | | PVA0.75$C_{opt}$ | |
| | | | SP | 2.79 | | 2.89 | | 4.89 | |

### 2.3. Preparing and Curing of Specimens

Nano-silica and micro-silica materials have been considered partial replacements for cement. Increasing the percentages of such materials in the mortar would lead to a decrease in the amount of cement content. In addition, since a control specimen was obtained with the workability of 155 mm, the superplasticizer was added in several steps in order to maintain the workability of the other mixture designs in the range of standard tolerance (±10 mm) until reaching the acceptable standard efficiency [38]. The superplasticizer content was considered a proportion of the cement weight, and it was achieved via a trial and error method with the workability of 155 ± 10 mm [38]. The percentage of PVA filaments was based on the percentage of mortar volume. Since the nano-silica was in the liquid phase, the percentage of solid materials was kept at 18% for the evaluation of water content in the mix designs, including nano-silica. The mixing scheme is depicted in Table 5.

The specimens were cubic, with the dimensions of 40 × 40 × 160 mm. The flow of the specimens was determined using the flow table test within the range of 140 to 160 mm. The tests mentioned in the preceding sections were conducted on the specimens after their ultimate strengths were obtained.

### 2.4. Compressive Test

After preparing and carrying out the samples and their operations, on the seventh and twenty-eighth days, the samples were taken out of the water for the compression and flexural tests. After drying, the compressive strength test was performed. Compressive loading was carried out via separate steps. The compressive strength test was performed according to the ACI 318 standard [40,41]. The compressive strength result of the specimens has been presented as $F_c$ in MPa.

**Table 5.** Mix design specifications.

| Design and Constituent | Sand (g) | Cement (g) | Nano-Silica Solid Material Weight (g) | Nano-Silica (g) | Micro-Silica (g) | PVA (g) | Water(g) | SP (g) |
|---|---|---|---|---|---|---|---|---|
| A, NS1 | 1350 | 445.5 | 4.5 | 25 | - | - | 204.5 | 2.4 |
| A, NS2 | 1350 | 441 | 9 | 50 | - | - | 184 | 3.15 |
| A, NS4 | 1350 | 432 | 18 | 100 | - | - | 143 | 5.04 |
| A, NS6 | 1350 | 423 | 27 | 150 | - | - | 102 | 7.52 |
| A, NS8 | 1350 | 414 | 36 | 200 | - | - | 61 | 10.98 |
| A, NS10 | 1350 | 405 | 45 | 250 | - | - | 20 | 16 |
| B, SF1 | 1350 | 445.5 | - | - | 4.5 | - | 225 | 0.81 |
| B, SF2 | 1350 | 441 | - | - | 9 | - | 225 | 1.22 |
| B, SF4 | 1350 | 432 | - | - | 18 | - | 225 | 1.4 |
| B, SF6 | 1350 | 423 | - | - | 27 | - | 225 | 1.49 |
| B, SF8 | 1350 | 414 | - | - | 36 | - | 225 | 1.62 |
| B, SF10 | 1350 | 405 | - | - | 45 | - | 225 | 1.71 |
| C, NS1SF9 | 1350 | 405 | 4.5 | 25 | 40.5 | - | 204.5 | 2.97 |
| C, NS2SF8 | 1350 | 405 | 9 | 50 | 36 | - | 184 | 3.2 |
| C, NS3SF7 | 1350 | 405 | 13.5 | 75 | 31.5 | - | 163.5 | 4.1 |
| C, NS4SF6 | 1350 | 405 | 18 | 100 | 27 | - | 143 | 6.08 |
| C, NS5SF5 | 1350 | 405 | 22.5 | 125 | 22.5 | - | 122.5 | 8.51 |
| C, NS6SF4 | 1350 | 405 | 27 | 150 | 18 | - | 102 | 11.12 |
| D, PVA0.25 | 1350 | 450 | - | - | - | 2.93 | 225 | 1.98 |
| D, PVA0.50 | 1350 | 450 | - | - | - | 5.85 | 225 | 8.01 |
| D, PVA0.75 | 1350 | 450 | - | - | - | 8.78 | 225 | 14.90 |
| E, PVA.0.5Aopt, (PVA0.50NS6) | 1350 | 423 | 27 | 150 | - | 5.85 | 102 | 11.03 |
| E, PVA.0.75Aopt, (PVA0.75NS6) | 1350 | 423 | 27 | 150 | - | 8.78 | 102 | 17.51 |
| F, PVA.0.5Bopt, (PVA0.50NF10) | 1350 | 405 | - | - | 45 | 5.85 | 225 | 14.99 |
| F, PVA.0.75Bopt, (PVA0.75NF10) | 1350 | 405 | - | - | 45 | 8.78 | 225 | 17.01 |
| G, PVA.0.5Copt, (PVA0.50NS2SF8) | 1350 | 405 | 9 | 50 | 36 | 5.85 | 184 | 13 |
| G, PVA.0.75Copt, (PVA0.75NS2SF8) | 1350 | 405 | 9 | 50 | 36 | 8.78 | 184 | 22 |

*2.5. Bending Test*

A three-point loading device was used to perform the bending test. The device was specified to perform the bending test on mortar specimens. The specimens were placed on a roller support, and the third roller was placed in the middle of the element. The length of the rollers was between 45 and 55 mm. The three-point bending test, according to the BS EN 1015-11 [42] standard, was utilized to specify the bending strength of specimens.

The bending strength (maximum tensile stress) is calculated as follows:

$$F_B = \frac{1.5 \times F \times L}{b^3} \tag{1}$$

where

$F_B$ is the bending strength in MPa;
$b$ is the dimensions of the square section of the prism in mm;
$F$ is the applied load on the flexural beam until fracture in N;
$L$ is the distance between the centers of the support and the load cell in mm.

## 3. Results and Discussion

### 3.1. Compressive and Bending Strength

Seven different mix designs have been considered for the test procedure. Each mix design consisted of seven combinations and three samples. Each combination was subjected to compressive and bending tests. The results of the bending and compressive strength tests of each mix design mode (single, binary, ternary), and also the average (Av) and standard deviation (SD) for the compressive and bending strengths of mortar containing nano-silica, micro-silica and PVA are presented in Tables 6–12 and Figures 2–8, respectively. The results were obtained from two types of specimens for each mix design for the 7-day and 28-day specimens.

**Table 6.** Compressive and bending strengths of mortar containing nano-silica.

| Sample | Nano-Silica (g) | Compressive Strength, Mpa | | | | Bending Strength, MPa | | | |
| | | 7 Days | | 28 Days | | 7 Days | | 28 Days | |
| | | Av. | SD. | Av. | SD. | Av. | SD. | Av. | SD. |
| CS | - | 38.59 | 0.27 | 43.75 | 0.31 | 8.23 | 0.06 | 9.14 | 0.07 |
| NS1 | 25 | 54.22 | 0.81 | 55.16 | 0.42 | 9.34 | 0.04 | 10.2 | 0.05 |
| NS2 | 50 | 55.31 | 1.04 | 56.09 | 0.85 | 10.13 | 0.03 | 10.37 | 0.02 |
| NS4 | 100 | 55.63 | 1.08 | 56.88 | 1.04 | 10.43 | 0.00 | 10.51 | 0.01 |
| NS6 | 150 | 57.5 | 0.44 | 60 | 0.50 | 10.76 | 0.02 | 11.03 | 0.03 |
| NS8 | 200 | 54.38 | 0.44 | 57.03 | 0.38 | 10.55 | 0.07 | 10.89 | 0.08 |
| NS10 | 250 | 53.44 | 0.31 | 56.41 | 0.32 | 10.14 | 0.17 | 10.3 | 0.11 |

**Table 7.** Compressive and bending strength of mortar containing micro-silica.

| Sample | Micro-Silica (g) | Compressive Strength, MPa | | | | Bending Strength, MPa | | | |
| | | 7 Days | | 28 Days | | 7 Days | | 28 Days | |
| | | Av. | SD. | Av. | SD. | Av. | SD. | Av. | SD. |
| CS | - | 38.59 | 0.27 | 43.75 | 0.31 | 8.23 | 0.06 | 9.14 | 0.07 |
| SF1 | 4.5 | 44.38 | 0.44 | 51.41 | 0.33 | 8.02 | 0.045 | 10.17 | 0.044 |
| SF2 | 9 | 50.47 | 0.27 | 57.03 | 0.17 | 9.00 | 0.02 | 10.46 | 0.025 |
| SF4 | 18 | 52.19 | 0.31 | 60.31 | 0.21 | 10.39 | 0.225 | 10.63 | 0.021 |
| SF6 | 27 | 55.00 | 0.45 | 63.91 | 0.32 | 10.41 | 0.075 | 10.90 | 0.095 |
| SF8 | 36 | 56.25 | 0.44 | 62.81 | 0.41 | 10.69 | 0.76 | 10.75 | 0.62 |
| SF10 | 45 | 57.97 | 0.53 | 64.69 | 0.48 | 10.76 | 0.05 | 11.61 | 0.061 |

**Table 8.** Compressive and bending strength of mortar containing micro-silica and nano-silica.

| Sample | Micro-Silica (g) | Nano-Silica (g) | Compressive Strength, MPa | | | | Bending Strength, MPa | | | |
|---|---|---|---|---|---|---|---|---|---|---|
| | | | 7 Days | | 28 Days | | 7 Days | | 28 Days | |
| | | | Av. | SD. | Av. | SD. | Av. | SD. | Av. | SD. |
| CS | - | - | 38.59 | 0.27 | 43.75 | 0.31 | 8.23 | 0.06 | 9.14 | 0.07 |
| NS1SF9 | 40.5 | 25 | 50.31 | 0.52 | 62.34 | 0.41 | 9.02 | 0.035 | 10.59 | 0.03 |
| NS2SF8 | 36 | 50 | 52.34 | 0.31 | 65.00 | 0.27 | 9.43 | 0.025 | 11.30 | 0.018 |
| NS3SF7 | 31.5 | 75 | 50.78 | 0.35 | 62.50 | 0.22 | 10.24 | 0.25 | 11.03 | 0.025 |
| NS4SF6 | 27 | 100 | 57.34 | 0.43 | 61.25 | 0.28 | 10.64 | 0.085 | 10.77 | 0.075 |
| NS5SF5 | 22.5 | 125 | 56.72 | 0.41 | 62.66 | 0.36 | 10.32 | 0.06 | 10.56 | 0.07 |

**Table 9.** Compressive and bending strengths of mortar containing PVA.

| Sample | PVA (g) | Compressive Strength, MPa | | | | Bending Strength, MPa | | | | Fracture Behavior Mode |
|---|---|---|---|---|---|---|---|---|---|---|
| | | 7 Days | | 28 Days | | 7 Days | | 28 Days | | |
| | | Av. | SD. | Av. | SD. | Av. | SD. | Av. | SD. | |
| CS | - | 38.59 | 0.27 | 43.75 | 0.31 | 8.23 | 0.06 | 9.14 | 0.07 | Brittle |
| PVA0.25 | 2.93 | 48.28 | 0.42 | 49.53 | 0.55 | 9.71 | 0.06 | 10.04 | 0.051 | Brittle |
| PVA0.50 | 5.85 | 33.91 | 0.46 | 40.94 | 0.38 | 8.07 | 0.07 | 7.79 | 0.056 | Bridge on the crack |
| PVA0.75 | 8.78 | 26.56 | 0.31 | 31.09 | 0.46 | 6.91 | 0.03 | 7.28 | 0.041 | Bridge on the crack |

**Table 10.** Compressive and bending strengths of mortar containing PVA and optimum percentages of NS.

| Sample | Nano-Silica (g) | PVA (g) | Compressive Strength, MPa | | | | Bending Strength, MPa | | | | Fracture Behavior Mode |
|---|---|---|---|---|---|---|---|---|---|---|---|
| | | | 7 Days | | 28 Days | | 7 Days | | 28 Days | | |
| | | | Av. | SD. | Av. | SD. | Av. | SD. | Av. | SD. | |
| CS | | - | 38.59 | 0.27 | 43.75 | 0.31 | 8.23 | 0.06 | 9.14 | 0.07 | Brittle |
| PVA0.50NS6 | 150 | 5.85 | 53.13 | 0.25 | 58.13 | 0.36 | 9.89 | 0.02 | 10.35 | 0.015 | Brittle |
| PVA0.75NS6 | 150 | 8.78 | 41.56 | 0.31 | 46.56 | 0.47 | 8.23 | 0.05 | 9.11 | 0.03 | Bridge on the crack |

**Table 11.** Compressive and bending strengths of mortar containing PVA and optimum percentages of SF.

| Sample | PVA (g) | SF (g) | Compressive Strength, MPa | | | | Bending Strength, MPa | | | | Fracture Behavior Mode |
|---|---|---|---|---|---|---|---|---|---|---|---|
| | | | 7 Days | | 28 Days | | 7 Days | | 28 Days | | |
| | | | Av. | SD. | Av. | SD. | Av. | SD. | Av. | SD. | |
| CS | - | - | 38.59 | 0.27 | 43.75 | 0.31 | 8.23 | 0.06 | 9.14 | 0.07 | Brittle |
| PVA0.50SF10 | 5.85 | 45 | 35.00 | 0.65 | 51.88 | 0.69 | 7.66 | 0.051 | 9.60 | 0.075 | Brittle |
| PVA0.75SF10 | 8.78 | 45 | 30.31 | 0.75 | 45.78 | 0.43 | 7.22 | 0.046 | 9.46 | 0.036 | Bridge on the crack |

**Table 12.** Compressive and bending strengths of mortar containing PVA and optimum percentages of NS and SF.

| Sample | PVA (g) | NS (g) | SF (g) | Compressive Strength, MPa | | | | Bending Strength, MPa | | | | Fracture Behavior Mode |
|---|---|---|---|---|---|---|---|---|---|---|---|---|
| | | | | 7 Days | | 28 Days | | 7 Days | | 28 Days | | |
| | | | | Av. | SD. | Av. | SD. | Av. | SD. | Av. | SD. | |
| CS | - | - | - | 38.59 | 0.27 | 43.75 | 0.31 | 8.23 | 0.06 | 9.14 | 0.07 | Brittle |
| PVA0.50NS2SF8 | 5.85 | 36 | 50 | 35.16 | 0.32 | 49.06 | 0.35 | 7.96 | 0.08 | 8.53 | 0.05 | Brittle |
| PVA0.75NS2SF8 | 8.78 | 36 | 50 | 37.03 | 0.28 | 46.09 | 0.42 | 8.02 | 0.07 | 8.51 | 0.06 | Brittle |

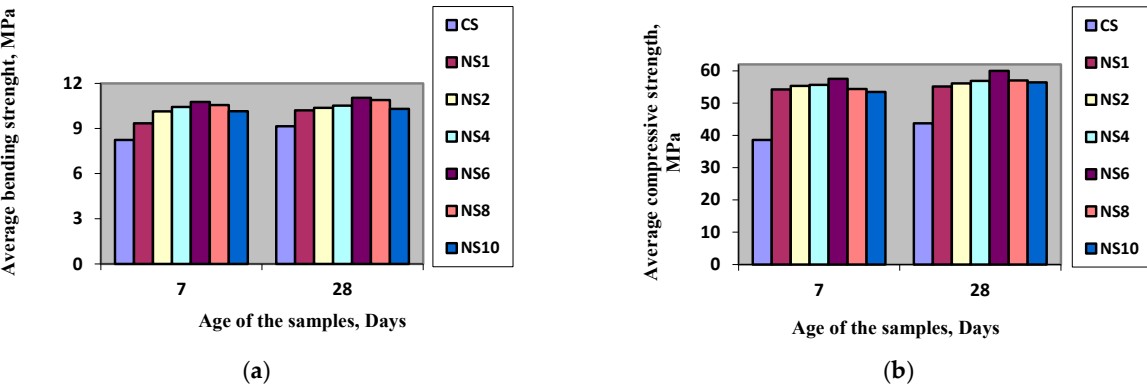

**Figure 2.** Compressive and bending strength of mortar containing Nano-silica.

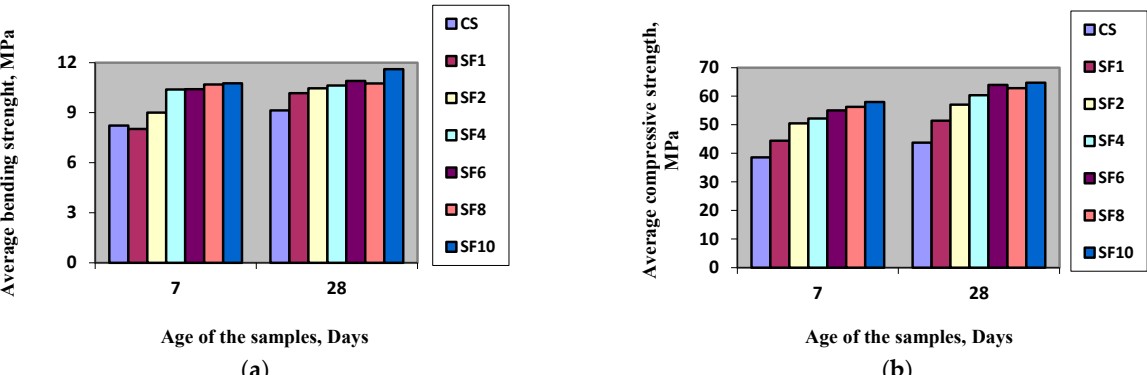

**Figure 3.** Compressive and bending strength of mortar containing micro-silica.

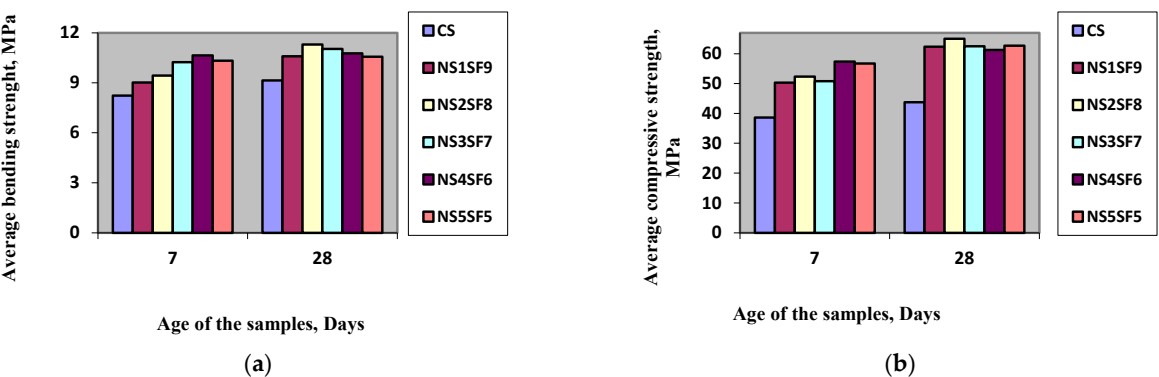

**Figure 4.** Compressive and bending strength of mortar containing nano-silica and micro-silica.

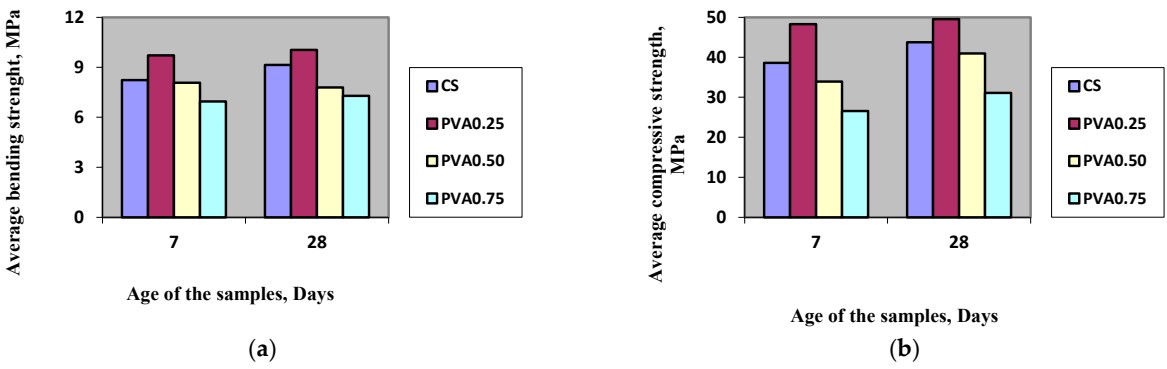

**Figure 5.** Compressive and bending strengths of mortar containing PVA fiber.

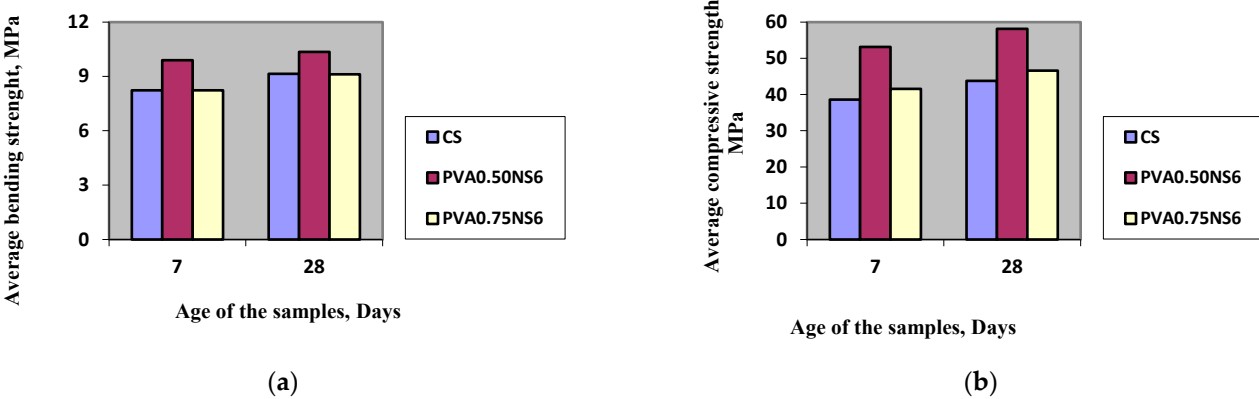

**Figure 6.** Compressive and bending strengths of mortar containing PVA fiber and optimum percentage of NS.

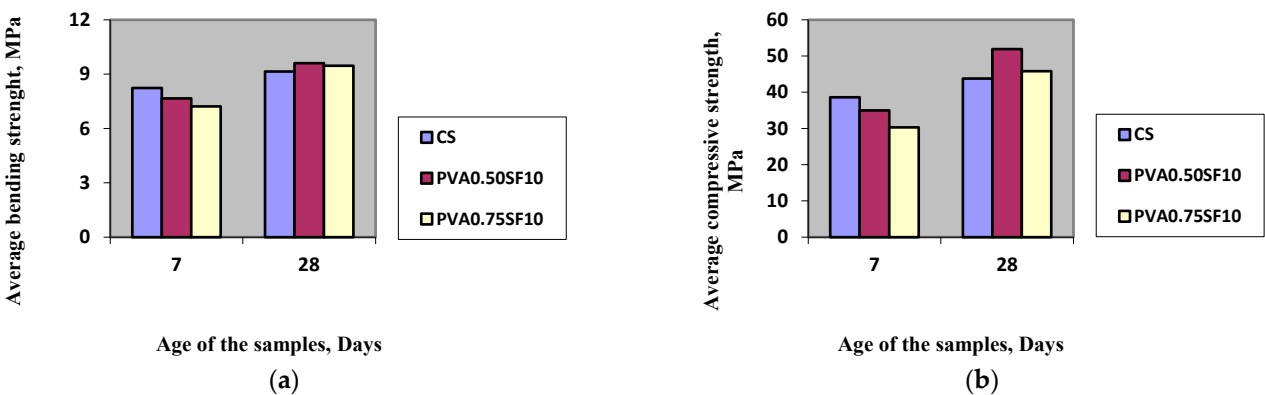

**Figure 7.** Compressive and bending strengths of mortar containing PVA fiber and optimum percentage of SF.

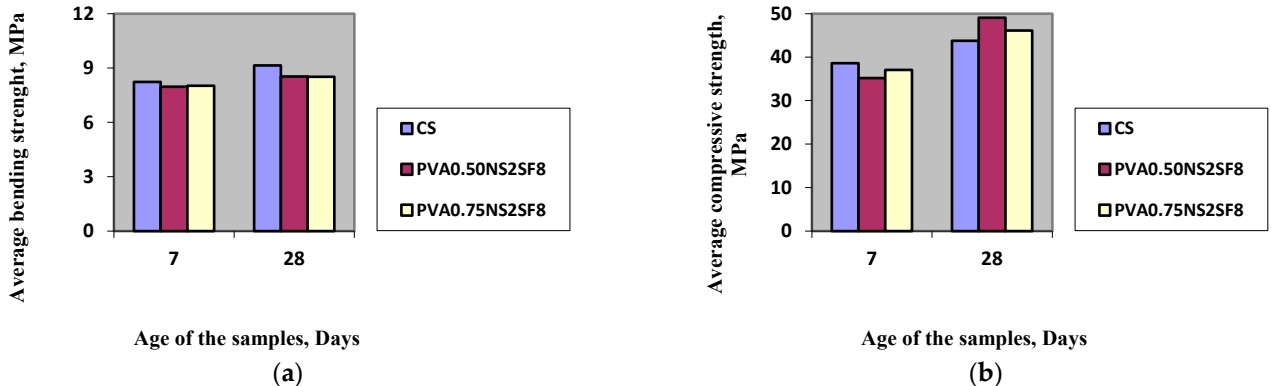

**Figure 8.** Compressive and bending strengths of mortar containing PVA fiber and optimum percentages of NS and SF.

For scheme A, the NS6 mix design was considered optimal, wherein the highest bending and compressive strengths were achieved—60 and 11.03 MPa for the 28-day specimens, respectively. It is worth mentioning that, by adding more nano-silica to the mixture, the bending and compressive strengths decreased. The highest bending and compressive strengths for Scheme B were observed in the SF10 mixture with 64.69 and 11.61 MPa for the 28-day specimens, respectively. The SF10 mix design was selected as the optimal one among the single-condition specimens for repairing purposes. For Scheme C, in which nano-silica and micro-silica (NS + SF) were included, the NS2SF8 displayed the

highest bending and compressive strengths at 65 and 11.30 MPa, respectively. It should be noted that, as the amount of nano-silica increased in the current scheme, the bending and compressive strengths decreased.

For Scheme D, although the highest bending and compressive strengths were obtained in the PVA0.25 mixture with 49.33 and 10.04 MPa, the phenomenon of bridging between cracks was not observed. However, this phenomenon was observed for the other mixtures in this scheme, with lower bending and compressive strengths.

For Scheme F, the highest bending and compressive strengths were achieved in the PVA0.5SF10 mix design, at 51.88 and 9.60 MPa for the 28-day specimens, respectively. A bridging phenomenon between the cracks (fracture mode) was not observed in this scheme; consequently, by considering the bridging phenomenon (ductile fracture), the PVA0.75SF10 mixture can be assumed to be the optimal design, with lower bending and compressive strengths.

For Scheme G (the ternary mixture), the highest bending and compressive strengths were associated with PVA0.5NS2SF8, at 49.06 and 8.53 MPa, respectively. However, the bridging phenomenon between the cracks was not seen in this mixture.

The SF10, the NS2SF8, and the PVA0.75SF10 mixtures can be considered the optimal mix designs in the single, binary and ternary modes, respectively. The compressive strength value of the NS2SF8 mix design, as the optimized mixture of the binary mode, was roughly 0.5% higher than that of the SF10 and approximately 29.5% higher than that of the PVA0.75SF10 mix design. The flexural strength of the NS2SF8, assumed to be the optimal compound, was approximately 2.74% lower than that of the SF10 optimized mixture and roughly 16.28% higher than that of the PVA0.75SF10 mixture.

For Schemes E, F, and G, the specimens prepared with 0.25 volumetric percent of PVA filaments presented much more brittle fracture behaviors because of the low percent of PVA filaments.

In terms of the mechanism of the nanoparticles and fibers leading to the improvement of mechanical characteristics, it is noteworthy that the main reason for the improvement is nucleation. According to the nucleation effect, based on the Centroplasm theory [43], nano-silica particles form nuclei for the development of cement hydration products in solution, which act as a bone for the cement gel, resulting in a reduction in the porosity of the mixture. Furthermore, it increases the speed of the reaction and the speed of the increases in compressive and flexural strengths.

In the following section, the results of the compressive and flexural strengths obtained via the testing procedure are compared to those from previous research works. Table 13 shows the results of the compressive and flexural strengths for the various mix designs in the current study and the results achieved in the previous investigations.

The results for all the mix designs are shown in the table below, except the following: NC (nano-CuO), NF (nano-$Fe_2O_3$), NA (nano-$Al_2O_3$).

### 3.2. Compatibility Test

A compatibility test was carried out for the optimized mixtures and the reference mortar to evaluate the properties of the repairing materials. If fractures occur in the repaired section or on a third of the girder, the optimized mixture and the initial concrete would be assumed to be compatible; otherwise, they would be incompatible [44]. A fracture occurred in the repaired sections in all the experiments with optimized mixtures, such as PVA0.75SF10, NS2SF8, and SF10; therefore, the compatibility of the optimized mixtures and the reference mortar was acceptable. The test setup for such an experiment is illustrated in Figure 9.

**Table 13.** Comparison of the compressive and flexural strength results of the current study with the previous research.

| Study | Control Specimen (MPa) | | Maximum Sample Strength (MPa) | |
|---|---|---|---|---|
| | Compressive Strength | Flexural Strength | Compressive Strength | Flexural Strength |
| 1　Current Study | 43.75 | 9.14 | NS6 → 60<br>SF10 → 64.69<br>PVA0.75 → 31.09<br>NS2SF8 → 65<br>PVA0.75SF10 → 45.38 | NS6 → 11.03<br>SF10 → 11.61<br>PVA0.75 → 7.28<br>NS2SF8 → 11.3<br>PVA0.75SF10 → 9.46 |
| 2　D. Alonso Dominguez (2017) | 58.5 | - | NS10 → 76.8<br>SF10 → 81.3 | - |
| 3　Ramezanian pour et al. (2017) | 48 | - | NS4 → 62 | - |
| 4　Chithra et al. (2016) | 56.5 | - | NS2 → 68.3 | - |
| 5　Mohseni et al. (2016) | 47 | - | NS5 → 57 | - |
| 6　Haruehansapong et al. (2014) | 23.52 | - | NS9 → 36.32 | - |
| 7　Madandoust et al. (2013) | 37.4 | 7.61 | NC3 → 43.53 | NF3 → 8.51 |
| 8　A.M. Said et al. (2012) | 60 | - | NS6 → 75 | - |
| 9　Nazari et al. (2011) | 31.6 | 4.2 | NS5 → 53.6 | NS4 → 6.9 |
| 10　Wan So et al. (2007) | 25.6 | - | NS12 → 68.8 | - |

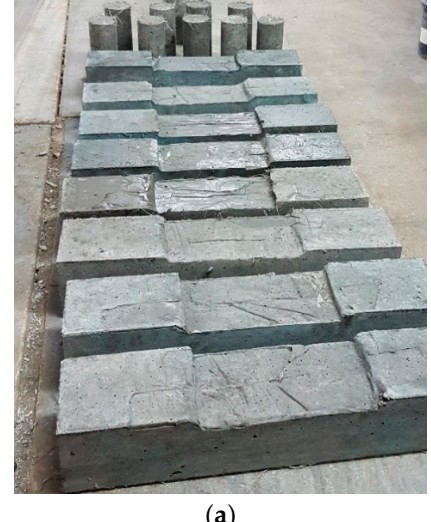

(**a**)

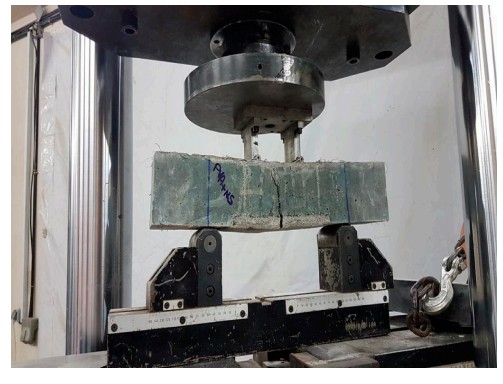

(**b**)

**Figure 9.** Compatibility of the specimens and the test setup [43]. (**a**) Preparation of specimens, (**b**) Test setup.

*3.3. Early Strength*

One of the most important properties of the cement-based repair mortars is the early increase in the strength of the mixture. In this regard, the compressive and bending strengths of the optimized mixtures, including PAV0.75SF10, NS2SF8, and SF10, were evaluated for the 7-day specimens and compared with the control specimens. The bending and compressive strengths of the SF10 optimized mix design increased by 50.2% and 30.7% compared to the control mixture, respectively. The bending and compressive strengths of the NS2SF8 optimized mixture increased by about 35.6% and 14.6% compared to the control specimen and decreased by 21.5% and 12.3%, respectively, in the PVA0.75SF10 mix

design. Consequently, the most optimal mixture as regards the early compressive and bending strengths was considered to be the SF10 mixture.

### 3.4. Efficiency

The strength increase rate coefficient ($\alpha$) is calculated according to Equation (2), based on the ACI 318 standard [41]. The efficiency parameter ($\alpha$) is shown in Table 14 for the SF10, NS2SF8, and PVA0.75SF10 mix designs, with 7-day and 28-day specimens, based on the obtained bending and compressive strengths.

$$\alpha = \frac{\Delta F}{\Delta T} \tag{2}$$

where

$\alpha$—the efficiency;
$\Delta F$—strength variations between the 7- and 28-day specimens;
$\Delta T$—time variations between 7- and 28-day specimens.

**Table 14.** The efficiency of the optimum mixtures.

| Efficiency | | Optimum Composition | Row |
|---|---|---|---|
| **Compressive Strength** | **Bending Strength** | | |
| 32% | 4% | SF10 | 1 |
| 60% | 9% | NS2SF8 | 2 |
| 74% | 11% | PVA0.75SF10 | 3 |

According to the table above, a noteworthy and novel result has been found. It can be concluded that the PVA0.75SF10 mix design requires the shortest curing period in comparison to other mix designs, including the reference mortar and other optimized mixtures.

### 3.5. Modulus of Rupture

The modulus of rupture parameter has been calculated for the cementitious mortar with various mix designs through a method similar to the one defined in ACI-318 for ordinary concrete. Thus, the $\beta$ coefficient is computed, which is defined as the ratio of the bending strength to the square root of the compressive strength (Equation (3)), and the results for all the 28-day mix designs have been investigated.

$$\beta = \frac{F_B}{\sqrt{F_c}} \tag{3}$$

In the supplementary file Tables S1–S3, and Figures S1–S9 are presented to further information about test setup and specimens.

### 4. Conclusions

This paper intends to study the mechanical characteristics of cement-based mortar containing nano-silica, micro-silica and PVA fiber. The silica particles and PVA fiber are added at various percentages in three different modes, including single, binary and ternary conditions. Various parameters, such as compressive strength, flexural strength, modulus of rupture, and efficiency, have been examined to determine the optimum mix design (with respect to the best improvement in mechanical characteristics) for each mixture condition.

In the binary state, in favor of the combination of silica-fume and nano-silica, the enhancement of mechanical characteristics was greater in comparison to the single state.

The PVA0.75SF10 mix design had the highest compressive and bending strengths, for which the bridging phenomenon between the cracks was evidently effective.

The maximum ratio of bending strength to the square root of the compressive strength (introduced as the β parameter for the evaluation of enhancement of mechanical characteristics) was estimated to reach the maximum ranges for the SF10, NS8, and PVA0.25 samples.

Finally, it is noteworthy to state that one would benefit from using nano-silica or a combination of nano-silica and micro-silica for the enhancement of strength. Hence, an effective ductile fracture mechanism would result from using PVA in the mix design.

**Supplementary Materials:** The following are available online at https://www.mdpi.com/article/10.3390/pr10091814/s1, Figure S1: Molding and curing of the specimens, Figure S2: The compression test setup, Figure S3: Three-point loading device to determine the bending strength (dimensions in mm). Figures S4–S9: SEM scans of PVA0.75SF10 NS2SF8 for various scales, Table S1: Mix design of the control mix design, Table S2: The β coefficient. Table S3: The β coefficient.

**Author Contributions:** Conceptualization, A.H., H.N.J., R.M., A.A. and S.A.H.; methodology, A.H., H.N.J., R.M., A.A. and S.A.H.; validation, A.H. and H.N.J.; investigation, A.H. and H.N.J.; resources, A.H. and H.N.J.; data curation, A.H. and H.N.J.; writing—original draft preparation, A.H., H.N.J. and R.M.; writing—review and editing, A.H., H.N.J. and R.M.; visualization, A.H. and H.N.J.; supervision, A.H.; project administration, A.H.; funding acquisition, A.H. All authors have read and agreed to the published version of the manuscript.

**Funding:** This research received no external funding.

**Conflicts of Interest:** The authors declare no conflict of interest.

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
