# Peer review of "Experimental Investigation on the Mechanical Characteristics of Cement-Based Mortar Containing Nano-Silica, Micro-Silica, and PVA Fiber"

_processes, doi:10.3390/pr10091814_

Round 1

Reviewer 1 Report (Previous Reviewer 3)

All the comments were addressed, I have no questions, and the paper can be published in its current form.

Author Response

As the reviewer stated, all the comments were addressed, I have no questions, and the paper can be published in its current form.

Reviewer 2 Report (Previous Reviewer 4)

The article by Jelodar et al. deals with cement based mortar with nanosilica, microsilica and PVA. The manuscript is written as a report more than a scientific article. Only the most important outcomes of the work should be presented and discussed, not all the experimental data obtained. There are 17 tables. That is too much for an article. Some tables should be removed or moved to the supplementary information.  The same happens with the figures. 12 it is too much. Some can be moved to the supplementary or removed.

This sentences are not understandable: One of the latest works could be addressed in which the relationships between the mechanical properties and loading speed of polypropylene fiber (PPF)-incorporated cement mortar at different ages, were studied [18].” What to address? Please, modify.

“Regarding interesting characteristics of nanomaterials, researchers investigated the effects of incorporating different types of such materials on integration of the concrete properties. “ which materials? Please modify.

I do not understand why the pH of fumed silica is given. Please explain.

Table 3 is not understandable. What is the chemical composition? In ionic nature it says ionic? Please modify

Table 7 is odd as well. Raw can be deleted. How a weight percentage can be expressed in gr?

The abbreviation for grams is g, not gr.

Conclusions are too odd. Should not be a list with points. Should summarize the work in a concise way highlighting the most important results.

 In addition, the English grammar should be checked by a native speaker. 

Taken into account the above considerations, I cannot recommed it for publication in the journal. 

Author Response

Reviewer 3 Report (Previous Reviewer 1)

The author has improved paper however there is no finding presented in the paper.

Author Response

Thanks a lot for valuable comments of the respectful reviewer.

The whole paper has been revised considering technical and writing matters.

The main achievement of the paper has been described in brief in the conclusion, and for the reviewers consideration it is presented again in the following:

This paper intends to study the mechanical characteristics of cement-based mortar containing nano-silica, micro-silica and PVA fiber. The silica particles and PVA fiber are added at various percentages in three different modes, including single, binary and ternary conditions. Various parameters, such as compressive strength, flexural strength, modulus of rupture, and efficiency, have been examined to determine the optimum mix design (with respect to the best improvement in mechanical characteristics) for each mixture condition.

In the binary state, in favor of the combination of silica-fume and nano-silica the enhancement of mechanical characteristics was more in comparison to the single state.

The PVA0.75SF10 mix design had the highest compressive and bending strengths, for which the bridging phenomenon between the cracks was evidently effective.

The maximum ratio of bending strength to the square root of the compressive strength (introduced as the β parameter for the evaluation of enhancement of mechanical characteristics) was estimated to reach the maximum ranges for the SF10, NS8, and PVA0.25 samples.

Eventually, it is noteworthy to state that one would benefit from using nano-silica or a combination of nano-silica and micro-silica for the enhancement of strength. Hence, an effective ductile fracture mechanism, would be resulted from using PVA in the mix design. 

Reviewer 4 Report (New Reviewer)

Dear Author, 

I am very happy to submit the review comments. The author worked a lot on this research. But I have mentioned a few of the review comments regarding this manuscript. Please rectify the below-mentioned review comments.

Review comments to the author (Processes-1847156)

In this manuscript, the author wrote an article entitled “Experimental investigation on the mechanical characteristics of cement-based mortar Containing Nano-Silica, Micro-Silica, and PVA fiber” suitable for publication, but the concerned author has to rectify the below-mentioned review comments in the “Processes”.

After rectifying the following reviewer’s comments, this article may accept. However, please rectify the following,

1.   To what extent does fiber-reinforced concrete rely on mechanical properties?

2.   You have used PVA fiber in your research, but you have not shown it anywhere in this manuscript.

3.   Just what kind of fiber is PVC?

4.   You have not shown the properties of nano-silica, micro-silica and particle size in this manuscript.

5.   When it comes to your composites' tensile properties, how will the stress-strain plot look?

6.   Need to include some comparative studies related to your work which is previously published.

7.   It would be best if you rectify the above comments and submit them once again for your expectation.

Round 2

Reviewer 2 Report (Previous Reviewer 4)

The article has been improved accordingly to the reviewers comments 

Author Response

Apparently the reviewer's comments has been responded.            

Reviewer 3 Report (Previous Reviewer 1)

The research idea has no originality.

Author Response

As responded to the respectful reviewer, the novelty of the current work has been reported in the conclusion. Moreover, it can be claimed that considering the combination of nano-silica and silica-fume (that is NS2SF8), which was one of the optimized mix designs, resulted to almost the largest value of flexural strengths among the previous works.

This manuscript is a resubmission of an earlier submission. The following is a list of the peer review reports and author responses from that submission.

Round 1

Reviewer 1 Report

The authors did not show originality. The manuscript does not make interest to readers as it does not raise significance or research finding.

Reviewer 2 Report

  • What is the novelty of the work?
  • Abstract:

    1. Avoid the use of abbreviations in the abstract.
    2. Well, the reader must at least understand the majority part of your work from the abstract instead of being confused and needing to read the manuscript to understand the meaning of your abstract.
  • Introduction:
    1. In line number 8: Specify the additional materials. If numerous materials are there then at least mentioned two to three materials.
    2. Inline number 17: Use of mineral admixture improves concrete properties. Which properties were found to be improved? Enlist the properties.
    3. Line 21-27, the author mentioned about work done in past by different researchers but not mentioned, that what was the observations made by these cited researchers. In addition to this conclusion drawn by cited researchers.
    4. The literature review is so vague, it is better to revise it.
  • Experimental Procedure:
    1. In line number 1: Specification is not the correct word here. It should be physical properties.
    2. Whether, the different properties of materials summarized in Table-1, Table-2, Table-3, and Table-4 evaluated in a lab or directly taken from the available specification?
    3. If the evaluated in lab, then which standards have been used? Mentioned the standards or respective codes.
    4. Fig.4 is cited in the main body of the text before Fig. 3. Please correct it.
    5. Replace Fig. 2, Fig. 3, and Fig. 4 which will highlight the main objective of the Figure. In addition to correct captions of these Figures.
    6. What is the size of Cube specimen tested under compression loading?
  • Results and Discussion:
    1. Table-8 to Table-14 is not cited in the main body of the text. Cite the mentioned table on their respective position.
    2. The results should be discussed more with the previous related study.
  • Conclusions:
    1. The conclusion section reads more like a summary. Please list key conclusions in this section.
  • General Comments:
    1. Check the formatting of main body text. At few places text font size is different.
    2. The quality of the figures should be improved.
    3. Pay attention to tables, figures, citations, and references.
    4. Check the plagiarism policy of the Journal.

Reviewer 3 Report

The structure is suitable, and the presented data and results are well described. The novelty of the work and the research gaps must be addressed in the introduction section. Also, add some more references discussing the working mechanisms of nano particles and fibers. These three references can help to discuss the working mechanism of nano particles.

  1. Effects of carbon nanotubes on expanded glass and silica aerogel based lightweight concrete, https://doi.org/10.1038/s41598-021-81665-y
  2. The Effect of Carbon Nanotubes on the Flowability, Mechanical, Microstructural and Durability Properties of Cementitious Composite: An Overview, https://doi.org/10.3390/su12208362
  3. Investigation on the Mechanical Properties and Post-Cracking Behavior of Polyolefin Fiber Reinforced Concrete, https://doi.org/10.3390/fib7010008

 "The following tests have been conducted on the specimens after gaining their ultimate strength",   remove the comma 

The authors mistakenly put the wrong figure numbering, "Figure 4. The compression test apparatus." needs to change to Figure 3. Similarly, figure 4 needs to be changed to figure 3. 

Discussion of the results is not sufficient. Please explain your results by citing previously published literature. Also, enhance the discussion by explaining the reason for the improvement or decline in strength. Explain the working mechanism of nano particles and fibres leading to improvement or decline in strength. Also, compare your results with previously published articles or put some data from literature in the discussion section. 

Authors showed similar results in the table and in the figures. It is better to choose only one way to present the data, either in tables or in figures.

Authors used different doses of nano particles and fibers and it would be nice to see some graphs showing the comparison of mechanical performances with the percentage of fibres and nano particles. 

Please mention how many samples of each type of concrete specimen were performed to be considered as final results. Also use an error bar for results. 

The conclusion is not well-written. Please rewrite it and focus on the main concept of the study. All conclusions must be convincing statements of what was found to be novel and impactful based on the strong support of the data, results, and discussion. The conclusion is rather lengthy and should be summarised further.

Reviewer 4 Report

The article by Jelodar et al deals with mechanical characteristics of cement-based mortar Containing Nano-Silica, Micro-Silica, and PVA fiber.

The topic is interesting for the journal. However, there is a previously published article with a similar topic: “Performance of Cement Mortar Composites Reinforced with Polyvinyl Alcohol Fibers” DOI:10.1088/1757-899X/518/2/022045

Therefore, I believe the novelty is not enough for publication in the journal.

Some other points:

-It says “The specifications for the PVA fiber are shown in Table 4 and Fig.1.”

The specifications are not in the figure.

-The article seems more a laboratory report than a scientific article, with many figures that hardly provide information (ie. equipment’s pictures) and little discussion, comparison with other authors, explanation of the results obtained, and so forth.  

The abstract is confusing, should be rewritten

The conclusions have many points. There should be concise in a single paragraph

SEM or TEM analysis of fractured specimens should be provided

Round 2

Reviewer 2 Report

In subsection 2.1

Are any guidelines followed for the selection of Type-I cement? What is the necessity to use only Type-I cement? Why not others?

Please check how to write pH?

To show the shape of the fiber please show with one single fiber. How to judge the shape by observing the whole bulk of fibers?

In subsection 2.2

Use subscript and superscript  wherever necessary

For sections 2.4 and 2.5

The content of subsections 2.4 and 2.5 is very short. Author needs to elaborate on the test description. Only showing the instrument in the Figure is not enough. Author need to elaborate on key points involved in the respective testing with their importance.

Figures 2 to 4 are very poor. Not fit for the quality of SCI journal.

General Comments:

Check the formatting of the main body text. At few places text font size is different.

The quality of the figures should be improved.

Pay attention to tables, figures, citations, and references.

Check the plagiarism policy of the Journal.

Reviewer 3 Report

Authors make corrections according to the reviewer comments. However some minor corrections still required before publication. 

Adhikary et al (2019) Showed that PVA have great influence on concrete strength, and self- compaction properties. Higher doses of PVA provide better flexural strength, and post-cracking behavior was also improved by the addition of fibers.

I think authors (adhikary et al.) didn’t used PVA, its polyolefin fiver. Please change the PVA  to polyolefin 

Section 3.1.1. Doesn’t need to be in separate section you can add it in 3.1. Just discus the literatures and the improvements in the discussion of compressive strength and bending strength. 

Reviewer 4 Report

I do thank the authors for their effort in replying my queries. However, my report is the same as previous one. There is not enough novelty for publication of the article in this journal. Besides, I was asking about SEM or TEM analysis of fractured specimens, not the nanoparticle alone.